# Neuroinflammation in Ischemic Stroke: Inhibition of cAMP-Specific Phosphodiesterases (PDEs) to the Rescue

**DOI:** 10.3390/biomedicines9070703

**Published:** 2021-06-22

**Authors:** Laura Ponsaerts, Lotte Alders, Melissa Schepers, Rúbia Maria Weffort de Oliveira, Jos Prickaerts, Tim Vanmierlo, Annelies Bronckaers

**Affiliations:** 1Biomedical Research Institute, Hasselt University, 3500 Hasselt, Belgium; laura.ponsaerts@student.uhasselt.be (L.P.); lotte.alders@uhasselt.be (L.A.); melissa.schepers@uhasselt.be (M.S.); 2European Graduate School of Neuroscience (EURON), Universiteitssingel 40, 6229 ER Maastricht, The Netherlands; jos.prickaerts@maastrichtuniversity.nl; 3Department Psychiatry and Neuropsychology, School for Mental Health and Neuroscience, European Graduate School of Neuroscience, Maastricht University, 6200 MD Maastricht, The Netherlands; 4Department of Pharmacology and Therapeutics, State University of Maringá, Maringá 87020-900, PR, Brazil; rubiaweffort@gmail.com

**Keywords:** ischemic stroke, neuroinflammation, neuroplasticity, PDE4, PDE7, PDE8, cAMP

## Abstract

Ischemic stroke is caused by a thromboembolic occlusion of a major cerebral artery, with the impaired blood flow triggering neuroinflammation and subsequent neuronal damage. Both the innate immune system (e.g., neutrophils, monocytes/macrophages) in the acute ischemic stroke phase and the adaptive immune system (e.g., T cells, B cells) in the chronic phase contribute to this neuroinflammatory process. Considering that the available therapeutic strategies are insufficiently successful, there is an urgent need for novel treatment options. It has been shown that increasing cAMP levels lowers neuroinflammation. By inhibiting cAMP-specific phosphodiesterases (PDEs), i.e., PDE4, 7, and 8, neuroinflammation can be tempered through elevating cAMP levels and, thereby, this can induce an improved functional recovery. This review discusses recent preclinical findings, clinical implications, and future perspectives of cAMP-specific PDE inhibition as a novel research interest for the treatment of ischemic stroke. In particular, PDE4 inhibition has been extensively studied, and is promising for the treatment of acute neuroinflammation following a stroke, whereas PDE7 and 8 inhibition more target the T cell component. In addition, more targeted PDE4 gene inhibition, or combined PDE4 and PDE7 or 8 inhibition, requires more extensive research.

## 1. Introduction

According to the World Stroke Organization, around 13.7 million individuals suffer a stroke each year [1]. With annually approximately 5.5 million deaths, stroke is the second leading cause of death in the world [1,2]. Moreover, stroke is considered a leading cause of morbidity, as up to half of stroke survivors remain permanently disabled [2,3]. Stroke can be recognized in patients through several clinical symptoms, including weakness/numbness of the face, leg or arm, dizziness, trouble with walking, loss of balance and/or coordination, confusion, and trouble with speaking or understanding [4]. Early detection of stroke symptoms is crucial since it can remarkably increase the chance of survival [4,5]. However, symptom detection is complicated by the fact that stroke symptoms can have an acute onset and are of transient nature, or symptoms can convert into a chronic state [6].

Stroke can be described as neurological deficits that arise after the rapid onset of a focal lesion in the central nervous system (CNS) with vascular origin [7,8]. Broadly, this disorder can be classified into two types: hemorrhagic and ischemic stroke, with the latter accounting for approximately 70–85% [1,9,10,11,12]. Considering that ischemic stroke is the most prevalent type, this review will focus on acute ischemic stroke.

Acute ischemic stroke manifests when a thrombus or embolus occludes a major cerebral artery, most commonly the middle cerebral artery [13,14]. The interrupted or severely impaired cerebral blood flow results in a series of deleterious events, called the ischemic cascade, which will eventually lead to irreversible neuronal damage [9]. The oxygen and nutrient deficiencies lead to an imbalance in energy demand and availability [9,14]. When neurons fail to sustain their transmembrane gradient because of disturbed ion and water homeostasis, neuronal electrical activity is compromised and functional deficits arise [7,9,10]. An increase of the extracellular glutamate concentration due to decreased glutamate reuptake and anoxic depolarization causes glutamate-induced excitotoxicity with massive neuronal calcium influx. Hence, the elevated intracellular calcium levels trigger the production of free radicals by neuronal nitric oxide synthase, culminating in oxidative stress and mitochondrial dysfunction (Figure 1). This induces apoptotic and necrotic cell death, resulting in massive neuronal cell death, leading to severe neurological damage [9,15].

In the ischemic core, neurons are irreversibly injured, causing loss of brain tissue and function within minutes [7,16]. The surrounding penumbra is a rim of mild to moderate ischemia characterized by a compromised electrical function, but due to collateral flow, cellular metabolism and viability are retained [16,17]. Spreading depolarizations originating in the ischemic core and propagating to the penumbral tissue increase the penumbral metabolic needs. If perfusion is not established within several hours, this tissue transitions into the necrotic core [9,10,16,17].

To date, only intravenous thrombolysis with tissue plasminogen activators (tPA) and endovascular therapy with a stent retriever (mechanical thrombectomy) have been approved as treatments of acute ischemic stroke [18,19]. These therapies cause recanalization to salvage penumbral tissue [13,20]. However, due to numerous contraindications (e.g., diabetes, use of oral anticoagulation) and the short therapeutic time window of 4.5–6 h, only 1–10% of acute ischemic stroke patients can be treated [18,21,22,23,24]. Moreover, many stroke patients, even when treated successfully, suffer from residual neurological impairments [2,3,25].

## 2. Neuroinflammation after Stroke

It is clear that the development of new, improved stroke therapies is of paramount importance. Preclinical research relies on the existence of several experimental stroke models to study its pathophysiology and novel therapeutics [27,28,29]. Nevertheless, the complexity and heterogeneous character of an ischemic stroke make mimicking all aspects of human stroke in a single animal model not feasible [27,28]. In general, multiple rodent stroke models exist to induce focal or global cerebral ischemic. Transient global ischemia can be elicited via cardiac arrest, bilateral carotid artery ligation, and four-vessel occlusion [29]. Nonetheless, focal cerebral ischemia models are preferred in stroke research, as they more closely resemble human stroke pathology. Several in vivo models exist to induce either transient (allowing reperfusion) or permanent focal ischemia, all bearing their advantages and disadvantages. These models include the intraluminal filament middle cerebral artery occlusion (MCAO) model, the craniotomy (distal) MCAO model, the photothrombotic model, the endothelin-1 model, and the embolic stroke model. A detailed discussion of these models can be found in reviews of Fluri et al. and Sommer et al. [27,28]. The transient or permanent MCAO model is considered as the in vivo tool that most closely simulates human ischemic stroke [30]. As this method to induce ischemic stroke generates well-reproducible lesions, it is the most commonly used method and has been applied in the majority of studies investigating neuroprotective and neuroinflammatory agents, as well as unraveling stroke pathophysiology. Another widely-used model is the photothrombotic model, producing thromboembolic clots, which is more appropriate when studying thrombolytic agents and pathophysiological processes after thrombolysis [27]

Over the years, there is a growing body of evidence supporting that attenuating neuroinflammation might be a valuable therapeutic target, as it is a major pathophysiological process implicated in stroke [31]. This key element of the multifactorial pathogenesis of ischemic stroke is initiated by different processes, including excitotoxicity, oxidative stress, release of damage-associated molecular patterns (DAMPs) from neural cells, and blood–brain barrier (BBB) dysfunction [9,30,31]. Suppression of neuroinflammation may prevent progressive destruction of cerebral tissue; thus, improving functional outcome [32]. Yet, the process of neuroinflammation is rather complex (Figure 2). Besides aggravating ischemic brain damage and compromising tissue viability, it also plays a beneficial role by promoting infarct resolution and recovery [22,30,33].

Microglia reside within the brain and are activated almost immediately after ischemic onset. They respond by secreting cytokines, chemokines, and matrix metalloproteinases [26,31]. As a result, leukocytes, mainly neutrophils, can infiltrate the ischemic brain via upregulation of cell adhesion molecules on endothelial cells [31]. Microglia play a dual role as they both secrete pro- and anti-inflammatory factors, resulting in, respectively, aggravated brain damage and infarct resolution [26]. In the acute phase, the inflammatory response elicited by these cells appears to contribute to ischemic pathology, while at a later stage they adopt a phagocytic phenotype for the removal of cellular debris, thereby contributing to infarct resolution [9].

After cerebral ischemia, astrocytes are activated by DAMPs, and exert both neurotoxic and neuroprotective effects. While glial scarring and the production of pro-inflammatory factors stimulate neurotoxicity, neuroprotection is mediated via angiogenic effects, neurotrophin release, and mitigating glutamate-induced excitotoxicity [33].

Stroke-induced disruption of the BBB allows the penetration of several peripheral immune cells, such as neutrophils, macrophages, natural killer cells, dendritic cells, T cells, and B cells into the ischemic lesion. These peripheral immune cells migrate towards the ischemic brain at different time points following stroke onset [33].

### 2.1. Innate Immune Cells

Starting from 30 min to several hours post-stroke, neutrophils are the first bloodborne immune cells to infiltrate the ischemic brain. Between days 1 and 3 after a stroke, neutrophil accumulation in the ischemic brain peaks, followed by a gradual decline over time. Neutrophils demolish the BBB and secrete pro-inflammatory factors, reactive oxygen species, and several proteases, all contributing to secondary injury [26,34]. In addition, their accumulation causes the ‘no-reflow’ phenomenon by obstructing blood flow [22]. Similar to microglia, neutrophils can adopt a phagocytic phenotype and this not only exerts neurotoxic, but also neuroprotective effects [9,22]. Neutrophil infiltration and accumulation appear to be of great interest for ischemic stroke patients. In animal models of stroke, ischemic injury is aggravated with high neutrophil infiltration, whereas neutrophil depletion lessens stroke disability [35].

Following neutrophil invasion, monocytes/macrophages infiltrate the stroked hemisphere, peaking at 3–7 days after the insult [26]. Macrophages can display and convert to multiple phenotypes depending on the ischemic environment and, thereby, exert different functions [36]. Initially, they display a pro-inflammatory phenotype and transition to an anti-inflammatory phenotype with lesion maturation [34]. Conversely, Hu et al. reported an increased expression of anti-inflammatory markers within the first 7 days post-stroke, followed by a decrease, while pro-inflammatory markers were persevered for at least 14 days after ischemia [37].

Natural killer cells appear in the ischemic brain rapidly after ischemic onset. They supposedly exhibit deleterious effects by stimulating neuroinflammation and neuronal cytotoxicity but this requires further research [34].

Under healthy conditions, dendritic cells are rarely present in the brain, but infiltration is observed in the first hours and up to 6 days after ischemic stroke. These cells recruit the adaptive immune system after presenting brain-derived antigens for recognition by T cells [33].

### 2.2. Adaptive Immune Cells

Post-stroke T cell invasion is observed after several hours and up to 30 days [38]. Within several hours after stroke onset, T cells cause thrombo-inflammation by facilitating endothelial adhesion of platelets and leukocytes. Accordingly, pro-inflammatory pathways are stimulated, aggravating brain damage. In contrast, T cell interactions with platelets may be beneficial as it prevents hemorrhagic transformation of ischemic stroke by exerting hemostatic effects [34]. Although T cell subsets exert numerous roles during the different stages after ischemic stroke, their precise mechanisms, and functions remain to be elucidated [38]. In general, CD8+ cytotoxic T cells, T helper 17 (Th17) cells, γδT cells, and Th1 cells are believed to exacerbate ischemic damage while regulatory T cells (Tregs) and Th2 cells exhibit neuroprotective effects [33,38]. Of all the T cell subsets, CD8+ cytotoxic T cells are the first to enter the ischemic brain. Activation of these cells is antigen-dependent, after which they induce neuronal cell death by releasing perforin and granzyme. Although the pro-inflammatory cytokine IL-17, secreted by γδT cells and Th17 cells in the acute phase, exacerbates brain damage, it promotes neurogenesis in later stages [33]. In addition, γδT cells and Th17 cells further aggravate brain injury by enhancing neutrophil activity and by modulating the FasL/PTPN2/TNF-α signaling cascade, thereby activating pro-inflammatory microglia. Activated CD4+ T cells alter the balance of microglial M1 (pro-inflammatory phenotype) and M2 (anti-inflammatory phenotype) polarization, thus aggravating neuronal death and neuroinflammation [38]. Differentiation of CD4+ T cells into either Th1 or Th2 cells is promoted by activated microglia and macrophages. Consequently, the secretion of pro-inflammatory (e.g., IL-2, IL-12, IFN-γ, TNF-α) or anti-inflammatory (e.g., IL-4, IL-10, IL-13) mediators ensures the neurotoxic and neuroprotective effect of Th1 and Th2 cells, respectively [33,38]. Moreover, while Th1 cells induce M1 microglial polarization, Th2 cells polarize microglia towards the healing M2 phenotype [38]. The immunomodulatory and immunosuppressive functions of Tregs convey neuroprotection and stimulate neural repair. The anti-inflammatory effect of Tregs is mediated by IL-10, which inhibits IL-1β and TNF-α [33,34]. Furthermore, via IL-10, Tregs favor the M2 microglial phenotype. In contrast, Tregs cause microvascular malfunction, compromising the integrity of the BBB, thus contributing to aggravated brain damage [33,39,40,41].

Whether B cells are beneficial or detrimental during the post-stroke recovery phase remains unclear. On one hand, they can produce anti-inflammatory cytokines, contributing to the attenuation of post-stroke inflammation. On the other hand, at a later stage, they cause brain damage, accompanied by neurological deficits via the development of antibodies against brain-derived antigens [34,42,43].

## 3. cAMP

The 3′-5′-cyclic adenosine monophosphate (cAMP) is a key player in multiple important biological processes, such as control of cellular differentiation in neuroinflammation [42]. Upon G-protein coupled receptor (GPCR) activation by their agonists such as β-adrenergic receptor agonists, the activated Gs protein activates adenylyl cyclase (AC) on the membrane, subsequently stimulating the removal of one pyrophosphate by a catalytic ATP, thereby forming cAMP [43]. Intracellular cAMP levels are tightly regulated on the subcellular level as they act as an important second messenger modulating a well-controlled wide range of physiological processes.

Multiple effector proteins mediate these cAMP-controlled biological functions including hyperpolarization-activated cyclic nucleotide regulated channels (HCN), cyclic nucleotide-gated channels (CNGC), exchange factor directly activated by cAMP (Epac), and protein kinase A (PKA). Both HCN and CNGC contain a cyclic nucleotide-binding domain (CNBD) and, therefore, can be directly regulated by the intracellular levels of cAMP. The binding of cAMP to this domain of HCN induces a conformational change and, thereby, increases the probability of non-selective ion channel opening during hyperpolarization [44], while binding to CNGC promotes specifically the influx of calcium [45]. Yet, the most known and studied effector mechanisms of cAMP signaling are the PKA-independent Epac and the PKA-dependent signaling pathways (Figure 3). Epac activation leads to axonal growth and neurotransmitter release in neurons, while in many other cell types this regulates proliferation, differentiation, adhesion properties and inflammatory responses [42]. Initiation of the PKA, referred to as classical cAMP signaling pathway, leads to phosphorylation of the most well-known downstream target of PKA, cAMP response element bound protein (CREB), leading to the increased transcription of genes containing an accessible CREB-binding site in their promotor (e.g., the somatostatin gene) [46,47,48,49]. By inducing the expression of the anti-apoptotic Bcl-2 and the neurotrophic growth factor BDNF, phosphorylation of CREB is often being described as neuroprotective and potentially angiogenic [50,51]. The biological effect of rising intracellular cAMP levels therefore highly depends on the downstream effector mechanisms being activated.

The neuroprotective feature of increasing intracellular cAMP levels has been described extensively over the past years. Activation of the PKA-CREB pathway by cAMP in neurons promotes neurite outgrowth and neuronal growth cone turning, leading subsequently to axonal growth and regeneration, both in vitro as in ischemic animal models [52,53,54]. Nevertheless, the role of intracellular cAMP signaling in glia and immune cells recently gained tremendous interest and is now considered a valuable target for effectively treating ischemic stroke. For instance, enhancing cAMP signaling in microglia skews the cells towards a repair-promoting and inflammation-reducing phenotype due to shifting the balance between the nuclear transcription factor NF-κβ, which signaling regulates the pro-inflammatory M1-like phenotype, to CREB signaling, which is required for regulating the anti-inflammatory M2 macrophages [55,56,57,58]. In astrocytes, increasing intracellular cAMP signaling attenuates post-ischemic nerve damage by regulating inflammatory cytokine release (including TNFα), increasing glutamate uptake and stabilizing the astrocytic membrane potential [59,60]. Yang et al. described that endothelial-mediated neuroprotective functions are enhanced when upregulating intracellular cAMP signaling as demonstrated by the increased trans-endothelial electrical resistance (TEER) and levels of the tight junction proteins claudin-5 and occludin [61]. Nevertheless, not only glial or brain microvasculature cells are affected by intracellular cAMP signaling, but also peripheral immune cells adapt their inflammatory properties. While cAMP-mediated PKA activation in neutrophils impairs their phagocytic capacity and MMP-9 secretion, the same effector mechanism activation reduces ROS activity and inflammatory cytokine secretion by monocytes/macrophages [62,63].

It has become clear that elevating cAMP levels can reduce peripheral immune activation and glia-mediated neuroinflammation, while exerting neuroprotective and neuroregenerative features. However, increasing cAMP levels through stimulation by growth factors or adenylyl cyclase activators are considered as generic forms of stimulation, thereby missing selectivity. Interestingly, to balance intracellular levels, cAMP is being metabolized by the enzymatic hydrolysis by members of the phosphodiesterase (PDE) family on the one hand, and the extracellular transport by multidrug resistance-associated proteins (MRPs) that actively extrude cAMP out of the cell on the other hand [64,65,66,67,68,69]. The PDE superfamily represents 11 families (PDE1-11), which are encoded by 21 genes (e.g., PDE4A-D). Each gene codes subsequently for multiple isoforms (e.g., PDE4B1-PDE4B5), resulting in over 72 different coding isoforms. PDE isoforms can be categorized based on their subcellular localization, tissue distribution, and substrate specificity (cAMP and/or cGMP) [70]. Besides the five dual substrate PDE families (PDE1, 2, 3, 10, and 11), three hydrolyze exclusively cGMP (PDE5, 6, and 9) and three exclusively cAMP (PDE4, 7, and 8) [70,71,72]. Therefore, to orchestrate different biological responses in the pathology of ischemic stroke, inhibition of PDE4, 7, or 8 might be considered as an interesting strategy for enhancing particularly intracellular cAMP signaling and subsequently attenuate neuroinflammation. By specifically inhibiting PDE genes or isoforms, intracellular cAMP levels can be increased in a more targeted manner. Considering the differential subcellular localization of the different PDE isoforms in specific cell types, this offers the opportunity of creating inhibition cAMP-specific PDE fingerprints tailored to these specific cell types. In this review, we discuss clinical implications, recent findings, and future perspectives regarding cAMP-specific PDE inhibition as a novel research interest for the treatment of ischemic stroke.

## 4. cAMP-Specific PDE Inhibition as Therapy for Neuroinflammation in Ischemic Stroke

### 4.1. PDE4 Inhibition

Several cAMP-specific pan-PDE4 inhibitors have been reported to improve ischemic stroke outcome by exerting anti-inflammatory, anti-apoptotic, and neuroplastic actions. PDE4 inhibitors include, among others, rolipram, roflumilast, FCPR03, and FCPR16 [55,73,74,75,76,77,78]. However, an important drawback of pan-PDE4 inhibitors is their dose-limiting emetic side effects [70]. These side effects are attributed to the stimulation of neurons in the area postrema together with the nucleus tractus solitarius of the brain stem and the nervus vagus. It has been reported that cAMP activates neurons in the area postrema, which in their turn stimulate the nervus vagus leading to nausea and emesis [79,80,81]. Thus, increasing cAMP levels in the brain generated by pan-PDE4 inhibitors can trigger emesis through activation of the area postrema. This effect is mainly attributed to inhibition of the PDE4D gene, which is highly expressed in the area postrema and the nucleus tractus solitarius [82,83,84]. To lower the stimulation of the emetic reflex, second-generation pan-PDE4 inhibitors with a lower emetic potential have been developed. These inhibitors include roflumilast, FCPR03, and FCPR16 [78,79,85]. The emetic potential of compounds can be investigated in rodents by means of the surrogate xylazine/ketamine test, and observation of emetic behavior in ferrets and dogs [85,86,87,88,89]. In the xylazine/ketamine test, the duration of anesthesia is indicative of a potential emetic effect of an administered compound. Compounds stimulating emesis will significantly lower the anesthesia time by activating α2-adrenoceptors, thereby acting as α2-adrenoceptors antagonists [85,86,87]. Vanmierlo et al. have shown by means of the xylazine/ketamine test that the emetic potential of the second-generation pan-PDE4 inhibitor roflumilast is 10 times lower compared to the first-generation pan-PDE4 inhibitor rolipram [86]. The emetic potential of FCPR03 has been investigated using both the xylazine/ketamine test in mice and observation of emetic behavior in beagle dogs. FCPR03 did not reduce the xylazine/ketamine-induced anesthesia time, indicating that FCPR03 does not evoke emetic side effects. In addition, beagle dogs receiving FCPR03 did not portray emetic behavior within a 180 min observation period [90]. Moreover, the pan-PDE4 inhibitor FCPR16 did not cause vomiting in beagle dogs up until 180 min after administration [77]. Another approach to circumvent emetic side effects besides the use of second-generation pan-PDE4 inhibitors (e.g., roflumilast), is the use of more specific PDE4 gene or isoform inhibition that are not involved in the emetic reflex (e.g., PDE4D inhibitors Gebr7b, Gebr32a, and BPN14770, PDE4B inhibitor A33) [91,92].

PDE4 inhibition is employed as an experimental treatment strategy in several different diseases because of its positive effects on (neuro)-inflammation, neuroplasticity, and cognition including memory [70]. Several studies have investigated the effect of pan-PDE4 inhibition following experimental ischemic stroke using a similar experimental set-up. Ischemic stroke was induced using the transient middle cerebral artery occlusion (tMCAO) model in either mice or rats. Animals were treated with different concentrations of pan-PDE4 inhibitors two hours after stroke induction by means of intraperitoneal injection (Table 1) [73,75,76,77]. Both first-generation (rolipram) and second-generation (roflumilast, FCPR03, FCPR16) pan-PDE4 inhibitors significantly reduced the lesion 24 h following stroke induction, while simultaneously improving neurological deficit scores in a dose-dependent manner [73,75,76,77]. In addition, Xu et al. have shown that treatment with FCPR03 improved the functional recovery in rats three days after stroke onset. FCPR03-treated rats performed significantly better in the behavioral tests, rotarod, and adhesive-removal test, where they spent more time on the rotarod and needed less time to remove the ipsilateral applied tape compared to the control group [76]. Furthermore, rolipram and FCPR16 reduced the neuroinflammation that takes place during stroke pathology [73,77]. Kraft et al. reported reduced expression of the pro-inflammatory cytokines IL-1β and TNF-α, with increased expression of the anti-inflammatory marker TGFβ1 [73]. In line with these results, Chen et al. demonstrated a significant reduction in the protein levels of TNF-α, IL-1β, and IL-6 [77]. The anti-neuroinflammatory effect of rolipram was partly attributed to reduced amounts of infiltrated neutrophils in the ischemic brain of rolipram-treated mice compared to vehicle-treated mice, 24 h after stroke induction. Pan-PDE4 inhibition also stabilized the BBB, shown by significantly decreased concentrations of Evans Blue, a vascular marker, leaking into the brain parenchyma. Along with less BBB leakage, expression of the tight junction protein claudin-5 was significantly upregulated in mice that received rolipram treatment compared to control mice that underwent a sham operation [73]. However, the positive effects of these studies on lesion size are contradicted by Yang et al. [74]. Yang et al. induced ischemic stroke in male rats utilizing the ligation model and the embolic model. Rats were treated with 3 mg/kg of rolipram either 30 min prior to stroke induction in the ligation model or 60 min before embolic experimental stroke induction (Table 1). No additional PDE4 inhibition treatment was administered following ischemic stroke onset. Yang et al. found the lesion sizes to be significantly increased following prophylactic rolipram treatment [74]. The observed negative effect of this prophylactic treatment strategy can potentially be explained by PDE4 compensatory mechanisms of increased activity of certain PDE4 genes or isoforms [74,93]. Previously, it has been reported that long-term pretreatment with rolipram induces a sustained increase of intracellular cAMP levels, consequently triggering a compensatory mechanism of increased PDE4 activity [93]. This increased PDE4 activity prior to experimental ischemic stroke might have led to lower cAMP levels, thereby worsening stroke outcome.

Studies by Bonato et al. and Soares et al. identified the effect of pan-PDE4 inhibition using roflumilast and rolipram on functional recovery following transient global cerebral ischemia (Table 1). Bonato et al. induced stroke in male Wistar rats and treated the animals with either 0.003 mg/kg or 0.01 mg/kg roflumilast, whereas Soares et al. applied C57Bl/6 mice that received either 0.1 mg/kg or 0.3 mg/kg rolipram. In both studies, treatment was initiated one hour following reperfusion by intraperitoneal injections and continued for 21 days [94,95]. The first-generation pan-PDE4 inhibitor rolipram was found to improve functional recovery in a dose-dependent manner by improved performance during the elevated zero maze, the object location task, and the forced swim test compared to vehicle-treated mice [95]. Roflumilast, a second-generation pan-PDE4 inhibitor, also improved functional recovery by alleviating spatial memory impairment compared to the vehicle treatment. In addition, roflumilast was shown to increase the levels of the anti-inflammatory cytokines IL-4 and IL-10, thereby contributing to a reduction in neuroinflammation [94].

Besides these in vivo results, in vitro analyses showed that both roflumilast and FCPR03 demonstrate anti-apoptotic properties in a mouse hippocampal neuronal cell line (HT-22 cells) and rat primary neurons. The involved mechanisms in this neuroprotection were identified as AKT/GSK3β signaling pathway activation and IRE1α/JNK pathway inhibition [75,76].

In connection with the previously mentioned PDE4 gene inhibition as a strategy to reduce emetic side effects, the PDE4B gene poses an interesting target. The PDE4B gene is abundantly expressed throughout the brain and in several inflammatory cells, including neutrophils, monocytes/macrophages, and microglia, as well as being involved in inflammatory reactions. In addition, increasing cAMP levels are known to exert anti-inflammatory effects [96]. To our knowledge, no research results are available yet on specific PDE4B gene inhibition in an ischemic stroke setting. However, this poses an interesting target for future stroke research considering its expression in the brain and involvement in the neuroinflammatory process. Furthermore, brain RNA sequencing results show that both the PDE4A and C genes are the least expressed PDE4 genes throughout the brain [97]. Even though PDE4A and PDE4C are also expressed in inflammatory cells including macrophages and microglia [96], their low brain expression patterns render them less interesting targets in neurodegenerative disorders. Interestingly, variations in the PDE4D gene have been associated with an increased risk of ischemic stroke even though conflicting results have been published on this manner. For example, in the Chinese population, both SNP83 and SNP87 of the PDE4D gene have been reported to confer a greater risk of developing ischemic stroke [98,99,100]. Several meta-analyses have been published that contradict these findings by stating that no genetic variants of the PDE4D gene can be significantly linked to ischemic stroke. Limitations identified by the meta-analyses themselves include an insufficiently large sample size to identify a significant association, and the suggestion that potentially existing associations are rather weak in nature [101,102,103]. In addition, it has been reported that the PDE4D gene is involved in ischemic stroke pathogenesis through differential expression of the PDE4D gene isoforms. Ischemic stroke patients were found to express reduced total PDE4D levels in their Epstein-Barr virus (EBV)-transformed B cell lines compared to healthy individuals because of significantly reduced expression of the PDE4D gene isoforms: PDE4D1, PDE4D2, and PDE4D5 [104].

In conclusion, both first-generation and second-generation pan-PDE4 inhibitors have led to positive effects in experimental ischemic stroke. More research is needed into specific PDE4 gene and isoform inhibition as a potential treatment strategy following ischemic stroke. In addition, more extensive research is needed to identify a potential significant association between PDE4D gene variants and ischemic stroke risk.

### 4.2. PDE7 Inhibition

PDE7A and B inhibition has been less extensively studied in a neuroinflammatory setting compared to PDE4 inhibition. Nonetheless, PDE7 inhibition is assumed to influence cell types of the innate immune system. Smith et al. reported expression of the PDE7 family, and more specifically the PDE7A gene, in neutrophils and monocytes, while studying chronic obstructive pulmonary disease (COPD) and asthma, diseases characterized by chronic inflammation [105]. However, upon investigating PDE7A protein levels, no expression was found in neutrophils. The lack of PDE7A protein expression in neutrophils could be explained by the lack of sensitivity of the employed Western blot analysis since more sensitive techniques, immunocytochemistry, and immuno-focal laser microscopy, did find PDE7A expression in neutrophils [105]. These results are relevant to ischemic stroke since neutrophils are the first peripheral immune cells to infiltrate the ischemic brain, closely followed by monocytes [33]. Additional research by Smith et al. confirmed PDE7A expression in monocytes, T cells, and tissue macrophages by means of Western blot analysis [106]. However, inhibition using the PDE7 inhibitor BRL150481 was found to have little effect on T cell proliferation and production of pro-inflammatory TNF-α by lipopolysaccharide (LPS)-stimulated monocytes and macrophages. These results show a low anti-inflammatory potential of PDE7 inhibition by BRL50481. Simultaneously, significant anti-inflammatory effects were observed upon treatment with a combination of BRL50481 and rolipram, a pan-PDE4 inhibitor. Interestingly, this suggests combined PDE4 and PDE7 inhibition as a potential therapeutic strategy for lowering neuroinflammation [106]. This could be particularly interesting as a strategy to reduce the dose of pan-PDE4 inhibitors as a means to circumvent their emetic side effects.

The adaptive immune system also plays a role in ischemic stroke pathogenesis, with T cells starting to infiltrate the brain after approximately one to two days. Thereby, T cells can worsen the inflammatory reaction and subsequent brain damage through production of pro-inflammatory cytokines, such as IL-2 [33]. Research has shown significant expression of the PDE7 family in T cells [107,108]. It has been demonstrated that increased PDE7 concentrations in T cells lead to decreased cAMP levels, accompanied by increased expression of IL-2. Since PDE7 inhibitors were not available at the time Li et al. conducted their research, PDE7 expression was blocked by PDE7 antisense oligonucleotides. Blockage of PDE7 reduced IL-2 expression and inhibited T cell proliferation [107]. These results are supported by research using the PDE7 inhibitor BC12, which also reduced IL-2 expression and T cell proliferation [108]. More targeted inhibition of the PDE7A gene using an inhibitor was also shown to decrease T cell proliferation and pro-inflammatory cytokine production [109]. Contradictory, another research group reported no significant alterations in T cell proliferation or cytokine production in PDE7A KO mice [110]. These contradicting results can potentially be explained by redundant presence of other PDEs in PDE7A KO mice, leading to apparent normal T cell function. In addition, there might be a difference in the expression of PDE7 in mouse and human T cells [109,110]. The previously mentioned studies are relevant to ischemic stroke research because it is suggested that PDE7 inhibition might be used as a therapeutic strategy modulating T cell activity. Therefore, it might be interesting to investigate whether PDE7 inhibition following ischemic stroke reduces neuroinflammation by mediating the T cell response.

PDE7 inhibition also exerts a neuroprotective and neuro-regenerative effect, and is expressed in neurons, astrocytes, and brain endothelial cells [97,111]. Morales-Garcia et al. have demonstrated increased neural stem cell proliferation and differentiation upon PDE7 inhibition in neural stem cell niches of the rat brain, namely the subgranular zone of the dentate gyrus and the subventricular zone. Additionally, PDE7 inhibition was found to stimulate neuronal cell formation in the dentate gyrus and olfactory bulb contributing to improved learning and memory behavior in rats [111].

To our knowledge, only one study is available so far on the effect of PDE7 inhibition on ischemic stroke outcome. Redondo et al. revealed that PDE7 inhibition significantly reduced nitrite production of LPS-stimulated primary astrocytes, microglia, and neurons in vitro. Furthermore, administration of PDE7 inhibitors significantly reduced infarct volumes accompanied by improved neurological outcome in mice 48 h after permanent MCAO induction [112]. These results provide the basis for a positive effect of PDE7 inhibition on ischemic stroke outcome. However, Redondo et al. did not investigate the in vivo potential anti-inflammatory effect of PDE7 inhibition treatment following ischemic stroke. More extensive in vivo research is thus needed to determine whether reduction of neuroinflammation and/or neuro-regeneration is the key mechanism by which PDE7 inhibition ameliorates ischemic stroke outcome.

### 4.3. PDE8 Inhibition

The PDE8 family is also classified as a cAMP-specific family of enzymes and its inhibition, like PDE7 inhibition, is less extensively studied in a neuroinflammatory setting compared to PDE4 inhibition. Human brain RNA sequencing results show PDE8A and PDE8B expression in neurons, oligodendrocytes, and endothelial cells [97]. Kobayashi et al. have shown that the PDE8B gene, especially, is abundantly expressed throughout the rodent brain [113].

PDE8 inhibition appears to affect inflammation through interaction with effector T cells [114,115,116,117,118]. It has been shown that PDE8 inhibition by dipyridamole leads to a reduced T cell migration in vitro towards the chemoattractant molecule CXCL12 [114]. PDE8 was also found expressed by endothelial cells. Dipyridamole treatment led to significantly increased cAMP levels in a mouse brain endothelial cell line, while simultaneously reducing expression of ICAM-1, VCAM-1, and CXCL1. In addition, PDE8 inhibition through dipyridamole was found to reduce αL and α4 integrins on effector T cells, which play an important role in T cell migration. Furthermore, Vang et al. also found that dipyridamole led to reduced adhesion of T cell blasts to mouse brain endothelial cells [115]. Thus, PDE8 inhibition affects T cell migration by reducing the CXCL12 production as well as T cell adhesion to endothelial cells. An important disadvantage of dipyridamole is its non-selective inhibition of PDE8 since it targets several PDE families, including PDEs 4–8, 10, and 11 [115,119,120,121]. It can thus be questioned whether the observed effects upon dipyridamole treatment are solely due to PDE8 inhibition or have to be attributed to combined PDE inhibition.

The PDE8-selective inhibitor PF-04957325 is available and has been reported to decrease adhesion of effector T cells to brain endothelial cells [117,118]. However, PF-04957325 treatment was demonstrated to not exert an effect on pro-inflammatory cytokine production or proliferation of T cells [118]. Interestingly, increased expression of PDE8 is seen in effector T cells compared to regulatory T cells [116,117]. Consequently, PDE8 inhibition only exhibited an effect on effector T cells by decreasing their adhesion to endothelial cells. In contrast to results found upon dipyridamole treatment, selective PDE8 inhibition was not found to alter expression of the T cell integrins, αL and α4 [117].

These results are of interest to ischemic stroke research because they show reduced interaction between T cells and brain endothelial cells. Potentially, PDE8 inhibition following ischemic stroke could reduce T cell infiltration into the ischemic brain through reduced interaction with the BBB, thereby reducing the neuroinflammatory reaction and subsequent brain damage. Extensive research into the potential effect of PDE8 inhibition on T cell—endothelial cell interaction in an ischemic stroke setting is needed before PDE8 inhibition can be considered as potential new therapeutic strategy.

In summary, inhibition of the cAMP-specific PDEs 4, 7, and 8 can be applied to ameliorate different stages in the pathophysiological process of ischemic stroke. PDE4 inhibition exerts a direct effect on the innate immunity, which plays an important role during the acute phase of ischemic stroke. Both PDE7 and 8 inhibition lower the T cell response, which comes into play during the chronic phase following stroke. By reducing both the innate and adaptive immune reactions, cAMP-specific PDE inhibition poses an interesting therapeutic strategy for lowering neuroinflammation following ischemic stroke. In addition, cAMP-specific inhibition can also be employed in the second phase of recovery following an ischemic stroke by stimulating neuroplasticity, thereby improving functional recovery.

### 4.4. Functional Read-Outs

Most research into new compounds as potential ischemic stroke therapeutic focuses on the short-term effect on lesion size and the neurological deficit score as an assessment of functional recovery. However, the Stroke Therapy Academic Industry Roundtable (STAIR) guidelines also advise long-term assessment of functional recovery [122,123]. Considering that ischemic stroke can cause impairment of cognition (such as attention, language, and memory), motor function, and sensorimotor function in patients, it is important to also assess these behaviors in experimental animal models in order to provide translational value on the investigated therapeutic to the human situation [8,124,125].

Several different behavioral tests are available to evaluate cognition and sensorimotor function following experimental ischemic stroke. Frequently, the applied cognitive tests include the Morris water maze, fear-conditioning test, and passive avoidance. Behavioral tests to assess sensorimotor function include the cylinder test, adhesive removal test, the rotarod, grip strength test, and the open field test (Table 2) [126,127]. The use of both sensorimotor and cognitive tests is recommended by the STAIR guidelines, as well as performing these behavioral assessments in a blinded randomized manner [123]. In addition, only a few studies investigate the effect on anxiety- and depressive-related behavior following ischemic stroke, even though many stroke patients also experience emotional impairments [124].

Although most studies focus on short-term improvement of rodent behavior following ischemic stroke induction, there are several studies available on the use of behavioral testing for the assessment of long-term functional outcome after experimental ischemic stroke, which is important to provide translational value of the research to the human situation [125,128,129]. Zhang et al. and Li et al. have demonstrated that the corner test is still sensitive at 42 and 90 days following ischemic stroke induction to assess sensorimotor function, thereby showing that the corner test is fit for long-term functional recovery assessment [125,128]. During the corner test, a mouse is placed between two boards that are positioned to form a corner with a small opening between the boards. When the mouse enters into the corner, its vibrissae are stimulated thereby making the mouse turn away from the corner. Mice that suffer from ischemic stroke will turn away from the corner with a preference towards the contralateral side (non-lesion side), whereas healthy animals will turn either way without showcasing a specific preference [125]. Bouët et al. have demonstrated the effectiveness of many behavioral tests to assess long-term functional recovery up until 26 days post-stroke, namely the adhesive removal test, the corner test, the passive avoidance test, and the staircase test [129]. Another behavioral test that can be employed for the long-term assessment of cognition, and more specifically memory, is the aversive eight-arm radial maze test. Bacarin et al. have demonstrated the use of the radial arm maze test up until 55 days following the induction of transient global cerebral ischemia [130]. In the aversive eight-arm radial maze test, a rat is placed on a circular platform to which eight arms are connected. The goal for the rat is to get through one of the arms into the goal box which is a darkened and closed space preferred by the rat over the open and light space of the circular platform and the arms of the maze. At the beginning of the test, all the arms in the maze are closed off, after some time all the arms are opened simultaneously, and the rat is allowed to explore the maze. When the rat enters an arm that contains a false goal box, the doors of the remaining arms will close simultaneously and the rat has to go back to the circular platform. At the end of each exploration period, the rat remains at the circular platform with closed-off arms for a certain period after which another exploration round starts. During the radial arm maze test, both learning and memory parameters are assessed as indicators of working memory [96,131,132].

When assessing rodent behavior following ischemic stroke, potential compensatory behavior should be taken into account. Both rats and mice are prone to developing compensatory behavior following injury, leading to better performance during behavioral tasks [126,133]. Previously, it has been suggested to employ several behavioral tests together with kinematic analysis to distinguish between true recovery and compensation [112].

In conclusion, future research into cAMP-specific PDE inhibition as potential therapeutic strategy following ischemic stroke should also include long-term assessment of the functional recovery following experimental ischemic stroke. By using behavioral tests to study functional recovery in terms of sensorimotor and cognitive impairments, increased translational value can be provided towards the human situation. In case cAMP-specific PDE inhibition is found to significantly improve long-term functional recovery in rodents, this might lead to a therapeutic providing improved quality of life for ischemic stroke patients.

### 4.5. Reflection on the Potential of PDE Inhibition in Other Neuroinflammatory-Related Disorders, Such as Traumatic Brain Injury, Alzheimer’s Disease, and Multiple Sclerosis

Reducing neuroinflammation by inhibiting cAMP-specific PDEs is not only an interesting therapeutic approach in ischemic stroke, but can also be employed in other neurodegenerative disorders characterized by neuroinflammation, such as traumatic brain injury (TBI), Alzheimer’s disease (AD), and multiple sclerosis (MS).

So far, TBI research has focused on inhibition of the PDE4 family as a potential therapeutic strategy. In animal models for TBI, decreased cAMP levels together with increased PDE4 levels have been reported [135,136]. PDE4 inhibition using rolipram has led to increased cAMP levels and a reduction in pro-inflammatory cytokine concentrations of TNFα and IL-1β [135]. More specific inhibition of PDE4B also reduced TNFα levels, while reducing neutrophil and microglia activation [143,144]. Furthermore, both pan-PDE4 inhibition and PDE4B inhibition have been shown to improve functional recovery following experimental TBI [136,143,144]. The results found in TBI research are relevant for ischemic stroke considering similar pathophysiology of both diseases and pan-PDE4 inhibition has rendered similar results in both diseases. Considering these similarities, PDE4B inhibition might also exert a similar effect following ischemic stroke since neutrophils and microglia are important players of the innate immune activation following stroke. This contributes to the rationale of researching more specific PDE4B inhibition as a strategy to reduce emetic side effects in an ischemic stroke experimental model.

Phosphodiesterases have also been suggested as a therapeutic target in AD [138]. Most research focusses on PDE4 inhibition as a means to lower the neuroinflammatory reaction involved in AD pathogenesis. In AD, the Aβ peptides are linked to aggravated neuroinflammation and upregulated PDE4B levels, leading to increased production of pro-inflammatory cytokines [139,145]. Subsequent PDE4B inhibition lowered the pro-inflammatory TNFα production by activated microglia [145]. In addition, pan-PDE4 inhibition also reduced the pro-inflammatory cytokine production of TNFα, IL-1β, iNOS, and NF-κβ [146]. Moreover, PDE4 inhibition has also been reported to enhance cognition and lead to neuroplasticity in AD as elaborated on in a review by Rombaut et al. [138,147]. The effect of pan-PDE4 inhibition in rodent models of AD is similar to the effect seen in ischemic stroke. The observed effects on cognition and neuroplasticity are interesting to ischemic stroke research since pan-PDE4 inhibitors might exert similar effects in the chronic phase of stroke pathophysiology thereby contributing to improved functional recovery in the later phase of the disease course.

Another neurodegenerative disorder characterized by a neuroinflammatory component is MS. Both PDE4 and PDE7 inhibition have been proposed as a potential therapeutic strategy for lowering neuroinflammation in MS patients [70,142]. PDE4 inhibition by rolipram treatment of experimental autoimmune encephalomyelitis (EAE) mice, an experimental model for MS, reduced neuroinflammation by decreasing immune cell infiltration into the CNS [148]. Rolipram treatment in EAE mice also led to reduced production of TNFα and IL-17, while simultaneously reducing T cell proliferation. The PDE7 inhibitor TC3.6 was found to exert a similar effect to rolipram by also reducing T cell proliferation and IL-17 production. Both rolipram and TC3.6 were reported to alleviate clinical EAE symptoms in mice [148]. In addition to an anti-neuroinflammatory effect in MS, pan-PDE4 inhibition has also been found to stimulate remyelination through an effect on oligodendrocyte precursor cell (OPC) differentiation, which is explained in more detail in the review by Schepers et al. [142]. In conclusion, the positive results of both PDE4 and PDE7 inhibition in MS are interesting to ischemic stroke research because they suggest that combined PDE4 and PDE7 inhibition can be employed as potential therapeutic strategy in ischemic stroke. By combining PDE4 and PDE7 inhibition, the dose of the pan-PDE4 inhibitor can be decreased thereby also reducing the chance of triggering the emetic reflex. In addition, also the remyelinating effect can be interesting in the chronic ischemic stroke phase to constitute an increased functional recovery in ischemic stroke patients.

## 5. Concluding Remarks

It has become clear that increasing cAMP levels through PDE inhibition can lower the neuroinflammatory reaction while stimulating neuroplasticity; hence, this is a promising therapeutic strategy for ischemic stroke. PDE4 inhibition targets the innate immune cells including neutrophils, monocytes/macrophages, and microglia during the acute ischemic stroke phase. On the other hand, both PDE7 and 8 pose interesting targets to temper the adaptive immune response that comes into play in during the chronic phase of disease. Hence, cAMP-specific PDE inhibition can be employed to reduce inflammation as well as to induce enhanced neurorehabilitation. However, caution should be taken when employing pan-PDE4 inhibitors because of their emetic side effects. Besides the use of second-generation pan-PDE4 inhibition with a lower emetic potential, these side effects could be circumvented by more specific PDE4 gene or isoform inhibition or by a combination of PDE4 with PDE7 or 8 inhibition.

Even though promising results were already obtained in both in vitro and in vivo stroke models, more extensive research is needed on the effect of cAMP-specific PDE inhibition in rodent stroke models in order to successfully translate the preclinical findings to a clinical application eventually. Future in vivo research should therefore also include several behavioral tests for the assessment of long-term functional recovery in order to provide a higher translational potential.

## Figures and Tables

**Figure 1 biomedicines-09-00703-f001:**
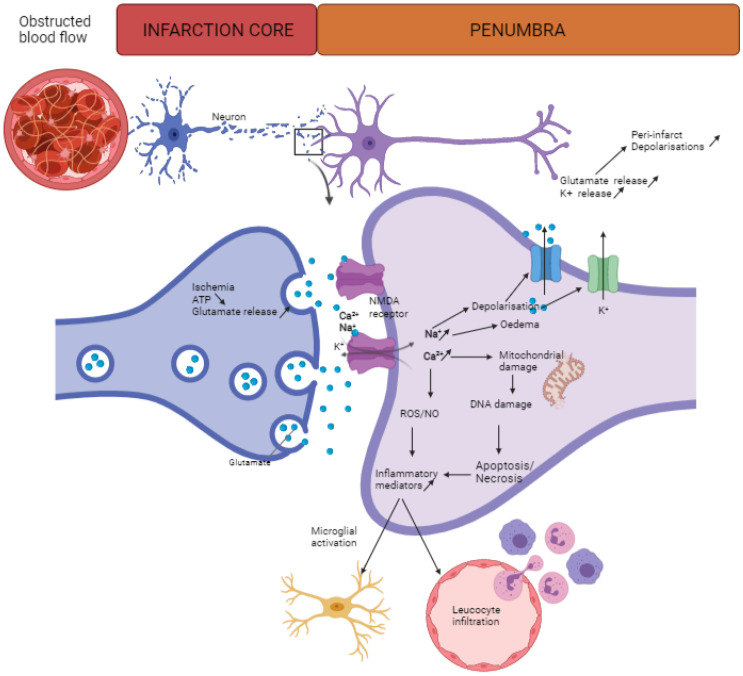
Overview of key events in the ischemic cascade: Ischemic stroke is caused by a sudden occlusion in the brain–blood supply. This results in irreversible damage of the neurons nearby the blood vessel, the ischemic core. This area is surrounded by the salvageable ischemic penumbra, in which neurons can be rescued if neuroprotective treatment is applied on time. In the core zone neurons, impaired oxygen and nutrient delivery results in a reduced ATP production, which will lead to loss of membrane potential and glutamate release. This will cause an ischemic cascade in neighboring neurons: the excess of glutamate will cause an increase in Na^+^ and Ca^2+^ influx, leading to cell swelling (edema) and depolarization, which will trigger K^+^ efflux and glutamate release. In addition, the increase in intracellular Ca^2+^ will also lead to mitochondrial damage as well as ROS and NO formation, both events causing mitochondrial damage and, thus, apoptosis and necrosis. This increased oxidative stress as well as the necrotic/apoptotic events lead to the secretion of inflammatory mediators, which exacerbate a negative effect of microglial activation and infiltration of native immune cells, such as neutrophils and macrophages [25,26]. Figure created with biorender.com (accessed on 1 May 2021).

**Figure 2 biomedicines-09-00703-f002:**
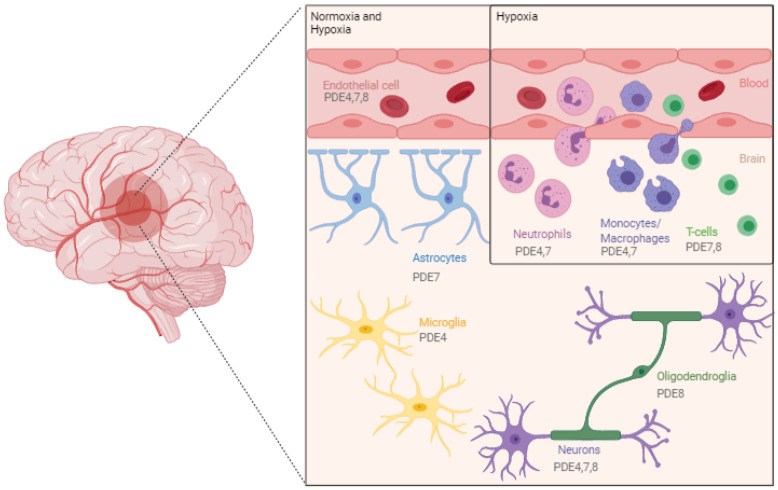
Overview of the most important cell types involved in the neuroinflammatory reaction of ischemic stroke pathophysiology and their expression of cAMP-specific phosphodiesterases (PDEs). Endothelial cells and neurons are found to express PDE4, 7, and 8, while astrocytes only express PDE7 and oligodendrocytes PDE8. PDE4 and 7 are both found in native immune cells, such as neutrophils and monocytes/macrophages, while T cells express PDE7 and 8. Microglia, which are considered the resident immune cells of the brain, possess PDE4. Figure created with biorender.com (accessed on 1 May 2021).

**Figure 3 biomedicines-09-00703-f003:**
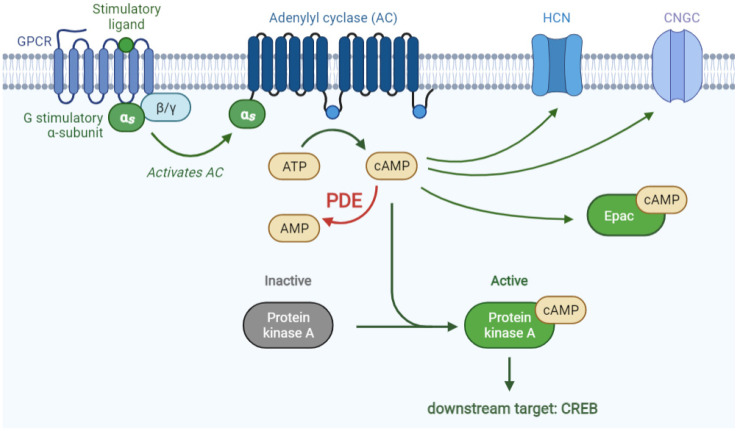
cAMP signaling pathway. Upon binding of a stimulatory ligand to a G-protein coupled receptor (GPCR), adenylyl cyclase (AC) is activated, subsequently stimulating the conversion of ATP to cAMP. cAMP can then activate its downstream targets including protein kinase A (PKA), hyperpolarization-activated cyclic nucleotide regulated channel (HCN), cyclic nucleotide-gated channel (CNGC), and exchange factor directly activated by cAMP (Epac). Activated PKA on its turn activates its downstream target cAMP response element bound protein (CREB). cAMP-specific phosphodiesterases (PDEs) catalyze cAMP to AMP. Figure created with biorender.com (accessed on 1 May 2021).

**Table 1 biomedicines-09-00703-t001:** In vivo studies researching pan-PDE4 inhibition in ischemic stroke. Abbreviations: I.P., intraperitoneal; NDS, neurological deficit score; EZM, elevated zero maze; OLT, object location task; FST, forced swim test. Symbols: ↑: increase; ↓: decrease.

Study	Compound	Concentration Inhibitor	Administration Route	Time Point of Treatment	Stroke Model	Mouse vs. Rat	Results
Kraft et al., 2013 [73]	Rolipram	2 mg/kg 10 mg/kg	I.P. injection	2 h post-stroke induction	tMCAO	Male C57Bl/6 mice	↓ lesion size Improved NDS ↓ neuroinflammation BBB stabilization
Yang et al., 2014 [74]	Rolipram	3 mg/kg	I.P. injection	30 min prior to stroke onset (1) 60 min prior to stroke onset (2)	(1) ligation model (2) embolic model	Male Fisher-344 rats Male Wistar rats PDE4D KO Fisher-344 rats	↑ lesion size
Xu et al., 2021 [75]	Roflumilast	0.3 mg/kg 1 mg/kg		2 h post-stroke induction	tMCAO	Male Sprague-Dawley rats	↓ lesion size Improved NDS
Xu et al., 2019 [76]	FCPR03	1.25 mg/kg 2.5 mg/kg 5 mg/kg		2 h post-stroke induction	tMCAO	Male Sprague-Dawley rats	↓ lesion size Improved NDS Improved functional recovery (rotarod, adhesive-removal test)
Chen et al., 2018 [77]	FCPR16	2.5 mg/kg 5 mg/kg 10 mg/kg	I.P. injection	2 h after ischemia	tMCAO	Male Sprague-Dawley rats Beagle dogs	↓ lesion size Improved NDS ↓ neuroinflammation No emesis induction
Bonato et al., 2021 [94]	Roflumilast	0.003 mg/kg 0.01 mg/kg	I.P injection	1 h after reperfusion (continued for 21 days)	Transient global cerebral ischemia	Male Wistar rats	Improved functional recovery (spatial memory) ↓ neuroinflammation
Soares et al., 2016 [95]	Rolipram	0.1 mg/kg 0.3 mg/kg	I.P. injection	1 h after reperfusion (continued for 21 days)	Transient global cerebral ischemia	C57Bl/6 mice	Improved functional recovery (EZM, OLT, FST)

**Table 2 biomedicines-09-00703-t002:** Frequently applied behavioral tests in rodents following experimental ischemic stroke.

Behavior	Behavioral Test	Stroke Model Applicability	Mice vs. Rats
Cognition (spatial learning)	Morris water maze	MCAO	Mice and rats [127,134]
Cognition (emotional memory and learning)	Fear-conditioning	MCAO	Mice and rats [129,135,136]
Cognition (memory)	Passive avoidance	MCAO	Mice and rats [127,129]
Cognition (spatial memory)	Aversive eight-arm radial maze	MCAO	Rats [55,130]
Sensorimotor function	Cylinder test	MCAO	Mice and rats [127,137]
Sensorimotor function	Adhesive removal test	MCAO	Mice and rats [127,131]
Sensorimotor function	Rotarod	MCAO (not in the photothrombotic model)	Mice and rats [129,138,139]
Sensorimotor function	Grip strength	MCAO	Mice and rats [127,132]
Sensorimotor function	Open field test	MCAO	Mice and rats [127,140]
Sensorimotor function	Corner test	MCAO	Mice and rats [127,141]
Motor function	Staircase test	MCAO	Mice and rats [129,137,142]

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
