# Peer review of "Neuroinflammation in Ischemic Stroke: Inhibition of cAMP-Specific Phosphodiesterases (PDEs) to the Rescue"

_biomedicines, 2021, doi:10.3390/biomedicines9070703_

Round 1
Reviewer 1 Report
Laura Ponsaerts et al., in the current review shed some light on the current knowledge and role of PDE inhibitors in context of Ischemic stroke. This review is timely and details the pressing needs on the development of more therapeutic options, potentially inhibitors of PDE4, 7 or 8. To my opinion, the readers of this review would benefit more if authors would provide more schematics to describe the mechanisms and pathways as mentioned below.
- Fig. 1 should also include the expression of PDEs in various cell types (including innate and adaptive immune cells)under physiological condition and compared side-by-side with stroke condition. So that readers will quickly get the whole picture on cell types involved with putative PDEs gene expression. Moreover, the Fig. seems not to be high resolution.
- Authors wrote couple of times regarding the effect of cAMP on the regeneration and brain plasticity hence improving the outcome of stroke. Again a signalling pathway by schematic would be very helpful.
- Under section 2 "Neuroinflammation after stroke" on Page 3, Lines 102-179, where authors describes the cell types and their roles in neuroinflammation. This section will be more clear by separating the innate cell types and adaptive cells under sub-headings.
- Reference is required on Page 2, Lines 84-85, where author mentioned "most closely simulates human ischemic stroke"
- Sentence correction is required on Page 3, Lines 108-109 where it states "at later stage they adopt...at a later stage.
- Correction required on Page 14, Line 634-637, this is duplication of paragraph from conflict of interest.
Author Response
Reviewer 1: Fig. 1 should also include the expression of PDEs in various cell types (including innate and adaptive immune cells) under physiological condition compared side-by-side with stroke condition. So that readers will quickly get the whole picture on cell types involved with putative PDEs gene expression. Moreover, the Fig. seems not to be in high resolution.
Authors: We thank the reviewer for this valuable remark. Figure 1 of the previous version of the manuscript has now become Figure 2, after inclusion of an extra figure as requested by reviewer 2.
The expression of PDEs can indeed depend on the presence of various stimuli in the micro-environment. For example, it has been shown that for in pulmonary smooth muscle cells PDE5 expression is increased after hyperoxia (Farrow et al., Circulation research, 2007). Expression and activity of distinct PDE4 isoforms (PDE4A and PDE4D) is augmented in response to hypoxia in eight different lung carcinoma cell lines (Pullamsetti et al., Oncogene, 2013). In microglia, it was found that pathological proteins amyloid-beta of Alzheimers’ disease induced increased expression of PDE4B (Zhang C et al. J Alzheimers Dis. 2014) and also various cytokines (such as TNF-α and IL-1β) increase expression of this isoform (Damien D. Pearse and Zoe A. Hughes, Glia, 2016).
While inhibition of several specific PDE genes and isoforms seems to hold anti-inflammatory and neuroprotective potential in ischemic stroke, the evidence of altered gene levels in specific cell types during ischemic stroke is scarce. While it can be postulated that it is clear that PDE4 levels increase upon neuroinflammatory stimuli in several cell types such as microglia and neutrophils, there is no knowledge on other PDE-subtypes and other cell types such as endothelial cells. Hence, the request of the reviewer to include in figure 1 all known changes of PDE expression in all cell types during stroke is impossible to fulfil. The reason why PDE inhibition has a great promise as ischemic stroke treatment is because of the various innate immune cells that invade the stroke lesion (neutrophils, monocytes and T-cells). To highlight that these cells are only present in the brain during stroke and thus stroke-associated neuroinflammation, we indicated in Figure 2 which cells are present after hypoxia. We hope that the reviewer and editor agree with this change.
The low resolution of the figure seems to be caused by the pdf-conversion.
Reviewer 1: Authors wrote a couple of times regarding the effect of cAMP on the regeneration and brain plasticity hence improving the outcome of stroke. Again a signalling pathway by schematic would be very helpful.
Authors: We have created Figure 3 on the cAMP signalling pathway to clarify the effects of cAMP.
Reviewer 1: Under section 2 “Neuroinflammation after stroke” on Page 3, Lines 102-179, where authors describe the cell types and their roles in neuroinflammation. This section will be more clear by separating the innate cell types and adaptive cells under sub-headings.
Authors: Subheaders “Innate immune cells” and “Adaptive immune cells” have been added in the section “Neuroinflammation after stroke”.
Reviewer 1: Reference is required on Page 2, Lines 84-85, where author mentioned “most closely simulates human ischemic stroke”
Authors: We have added the reference McBride et al. ‘Precision stroke animal models: the permanent MCAO model should be the primary model, not transient MCAO’, in which authors discuss the prevalence of strokes and what animal models most closely resemble different subtypes of stroke.
Reviewer 1: Sentence correction is required on Page 3, Lines 108-109 where it states “at a later stage they adopt … at a later stage.
Authors: This sentence has been corrected to: “In the acute phase, the inflammatory response elicited by these cells appears to contribute to ischemic pathology, while at a later stage, they adopt a phagocytic phenotype for the removal of cellular debris, thereby contributing to infarct resolution”.
Reviewer 1: Correction required on Page 14, Line 634-637, this is a duplication of paragraph from conflict of interest.
Authors: This duplication has been removed, this paragraph can now only be found under the section conflict of interest.
Reviewer 2 Report
Very well written review, it was delight to read. Discussion is well written addressing several important issues.
Please see few comments below. I believe that those changes will improve the quality and readability of the review and will help readers to better understand the underlying mechanism.
-Please describe in introduction symptoms patients experience during and after stroke. How long patients experience the symptoms?
Line 40-53:
-Please include a simplified Figure describing the events during acute ischemic stroke and the mechanisms involved. This will help a reader that is not so familiar with those events to better understand.
-Please include figure describing putative mechanisms of cAMP effects on stroke via PDE inhibition
-Please refer to figure 1 in text
Author Response
Reviewer 2: Please describe in introduction symptoms patients experience during and after stroke. How long patients experience the symptoms?
Authors: We have added stroke symptoms in the introduction, as well as information on how long symptoms can last. For this, the following sentences have been added to the manuscript: “Stroke can be recognized in patients through several clinical symptoms including weakness/numbness of the face, leg or arm, dizziness, trouble with walking; loss of balance and/or coordination, confusion, and trouble with speaking or understanding. Early detection of stroke symptoms is crucial since it can remarkably increase the chance of survival. However, symptom detection is complicated by the fact that stroke symptoms can have an acute onset and are of transient nature, or symptoms can convert into a chronic state.
Reviewer 2: Line 40-53, Please include a simplified figure describing the events during acute ischemic stroke and the mechanisms involved. This will help a reader that is not so familiar with those events to better understand.
Authors: We have added Figure 1 illustrating the events that take place starting from the obstruction of the blood flow, in the infarction core and the surrounding penumbra. This figure includes the involved mechanisms from impaired oxygen and nutrients supply to microglial activation and leukocyte infiltration.
Reviewer 2: Please include figure describing putative mechanisms of cAMP effects on stroke via PDE inhibition
Authors: This comment was also given by reviewer 1 and we have added Figure 3 illustrating the cAMP signalling pathway in which we show how cAMP is influenced by PDEs and its downstream effects.
Reviewer 2: Please refer to figure 1 in text.
Authors: We have included an in text reference to this figure. Figure 1 of the previous version of the manuscript has now become Figure 2, after inclusion of an extra figure on the stroke pathophysiology as requested (by this reviewer).